# First-Decade Biomass and Carbon Accumulation, and Woody Community Change after Severe Wind Damage in a Hemlock-White Pine Forest Remnant

Chris J. Peterson

Department of Plant Biology, University of Georgia, Athens, GA 30602, USA; chris@plantbio.uga.edu;
Tel.: +1-706-542-3754

**Abstract:** Studies of biomass and carbon dynamics and community composition change after forest wind disturbance have predominantly examined trends after low and intermediate severity events, while studies after very severe wind disturbance have been few. This study documents trends in aboveground biomass and carbon across 10 years of forest recovery after severe wind disturbance. In July 1989, a tornado struck mature *Tsuga canadensis-Pinus strobus* forest in northwest Connecticut, USA, causing damage across roughly 8 ha. Canopy tree damage and regeneration were surveyed in 1991 and 1999. Two primary hypotheses were tested, both of which derive from regeneration being primarily via the release of suppressed saplings rather than new seedling establishment: (1) Biomass and carbon accumulation will be faster than accumulation reported from a similar forest that experienced similar severity of wind disturbance; and (2) The stand will experience very little change in species composition or diversity. Estimated immediate post-disturbance (1989) aboveground live-tree carbon was $20.7 \pm 43.9$ Mg ha$^{-1}$ (9.9% of pre-disturbance) Ten years after the disturbance, carbon in aboveground live tree biomass increased to $37.1 \pm 47.9$ Mg ha$^{-1}$; thus for the first decade, annual accumulation averaged 1.6 Mg ha$^{-1}$ of carbon; this was significantly faster than the rate reported in a similar forest that experienced similar severity of wind disturbance. The species diversity of woody stems ten years after the disturbance was significantly higher (nonoverlapping confidence intervals of rarefaction curves) than pre-disturbance canopy trees. Thus, hypothesis 1 was confirmed while hypothesis 2 was rejected. This study augments the limited number of longer-term empirical studies that report biomass and carbon accumulation rates after wind disturbance, and can therefore serve as a benchmark for mechanistic and modeling research.

**Keywords:** *Betula*; disturbance; Connecticut; forest regeneration; *Prunus*; tornado; *Tsuga*; windthrow

## 1. Introduction

Disturbances have the potential to greatly alter community characteristics such as species composition and diversity, as well as ecosystem traits such as biomass and carbon pools [1,2]; but we have yet to achieve an understanding that fully addresses the interacting effects of type and severity of disturbance, forest characteristics, and abiotic and edaphic conditions. Development of such understanding surely requires empirical sampling from the full range of disturbance types and severities. Carbon dynamics have been documented predominantly after fire, harvest, and insect outbreak [3,4], while fewer examples are available after wind disturbances. In addition, the research available after wind disturbances has often followed low and intermediate severity events [5–7]. Such limitations hinder the development of robust theory and generalizations [8] that will enable useful predictions of how carbon dynamics will be altered from ongoing climate change that will likely bring altered intensity, size, and frequency of storms.

At the community level, substantial literature has developed over the last few decades describing post-wind damage species composition and diversity, see reviews in [9,10]. An especially noteworthy general trend is that wind disturbance to canopy trees often

results in the release of suppressed advance regeneration, and if the canopy disturbance is not too severe, survival and subsequent sprouting of canopy trees (lower severity events in [7]). When canopy-size survivors dominate the recovering forest e.g., [5,11], the compositional change from pre-disturbance to recovery is limited; whereas if released advanced regeneration becomes dominant, the compositional change depends on whether the advanced regeneration is made up of the same species as was in the pre-disturbance canopy e.g., [12,13], or if it is a different suite of species e.g., [14]. After an especially severe wind disturbance, abundant new seedling establishment may drive a shift in species composition (e.g., [15–17], high-severity events in [7]), usually towards a more early- or mid-successional suite of species. Thus, in terms of change in species composition, wind disturbance that damages canopy tree crowns while leaving lower trunks and root systems intact will result in the least change in species composition, while situations that result in an abundant new seedling establishment (e.g., very high severity, sometimes combined with chronic pre-disturbance herbivory that removes advance regeneration) will result in the most compositional change. Situations that result in regeneration from the release of advanced regeneration and subcanopy saplings, will result in limited compositional change if the advanced regeneration is populated by the same species as was in the canopy, or substantial compositional change if the advanced regeneration is compositionally distinct from the canopy species. Because this site had advanced regeneration comprised of similar species as those in the canopy, the first hypothesis tested here is that regeneration will show little change from pre-disturbance canopy tree composition or species diversity.

Trends observed in the literature allow the formulation of a general hypothesis regarding rates of biomass and carbon accumulation after wind disturbance. Perhaps the longest time span that reports biomass and carbon accumulation is [18]—71 years after a 1938 hurricane severely damaged the old-growth white pine forest remnant at the Harvard Tract of southern New Hampshire, they found aboveground carbon accumulation rates in live trees of 2.0 Mg ha$^{-1}$ yr$^{-1}$ (derived from biomass accumulation of 4.0 Mg ha$^{-1}$ yr$^{-1}$) in the first few years, decreasing to 0.6 Mg ha$^{-1}$ yr$^{-1}$ in the last few years. In an old-growth hemlock-beech forest of northwestern Pennsylvania, aboveground carbon accumulated in live trees at rates of 1–2 Mg ha$^{-1}$ yr$^{-1}$ during the first 20 years after especially severe wind damage [17]. Two accumulation studies in the Neotropics report slightly higher or similar accumulation rates. Following a hurricane in Nicaragua, rates of aboveground carbon accumulation of 2.3–2.8 Mg ha$^{-1}$ yr$^{-1}$ were reported [5] during the first twelve years; while Amazon forests damaged by squall-line storm events amassed aboveground carbon at mean net rates of 2 Mg ha$^{-1}$ yr$^{-1}$ [7] during years 4–27 post-disturbance. The Amazon trends are net biomass increment-biomass accumulation by tree and sapling growth was sometimes offset by highly-variable post-windthrow delayed mortality.

In summary, the rate of accumulation of biomass and carbon are expected to decrease across regeneration mechanisms in the order (fastest to slowest) of sprouting and refoliation of surviving adult stems, to the release of subcanopy and advanced regeneration, to germination of new seedlings. The site reported here had a substantial subcanopy and sapling layer that survived the storm and was subsequently released cf. [14,19,20]; therefore, the second hypothesis tested here is that first-decade biomass and carbon accumulation rates in this site will be higher than the rates reported in [17] where regeneration began with small seedlings.

## 2. Materials and Methods

### 2.1. Study Site and Disturbance Event

This study was carried out at Cathedral Pines, a 16.2 ha forest preserve owned by the Connecticut Nature Conservancy, and bordered on three sides by agricultural land, and on a fourth side by younger forest. It is near the village of Cornwall, Litchfield County, Connecticut (41°50′ N, 73°20′ W). Elevation ranges from roughly 200 to 250 m, slopes are up to 25°, and the aspect is westerly. Soils are Enfield silt loams (Entic Haplorthods and Aquic Fluventic Eutrochrepts, derived from numerous types of acid rocks) in lower positions

on the hillside, and Charlton very stony fine sandy loam (also an Entic Haplorthod, from schists) in higher elevations [21].

The average annual temperature at Falls Village NWS station, 15 km away, is 8.6 °C, the average July temperature is 21.2 °C, and the average January temperature is −4.6 °C. The mean length of the frost-free period is 141 days; the mean annual precipitation is 1042 mm [22].

The history of the Cathedral Pines stand is complex, but most of the dominant pre-windthrow pines date from a period from roughly 1790–1803, and the hemlocks from the mid-1800s [23,24]. The uniformity of pine ages and the presence of charcoal in a majority of sample pits around the site suggest that the site was cleared and burned in the mid-to-late 18th century, which allowed the white pines that were emergent trees in 1989 to establish [25]. Pre-disturbance vegetation was detailed in [26], and consisted primarily of mature individuals of hemlock (*Tsuga canadensis* Carr.) and white pine (*Pinus strobus* L.), with smaller proportions of white oak (*Quercus alba* L.), red maple (*Acer rubrum* L.), yellow birch (*Betula alleghaniensis* Ehrh.), and black birch (*Betula lenta* L.). The woody understory was well-developed and consisted mostly of black birches and small hemlocks. The herbaceous vegetation was sparse. Nomenclature follows [27].

On 10 July, 1989, an F2-rated tornado formed over Litchfield County, and moved south-southeast for 16 km in a series of hops, creating three separate swaths of destruction averaging 75 m wide [28]. In this area, the intensity of the tornado may have been as high as F3, with estimated wind speeds of 254–332 km/h (C. MacClintock, pers. comm.). Several reports attest that the winds caused nearly complete (>90%) canopy tree destruction in the northern and western half of the forest preserve, where my sampling was conducted [23,25,29]. Following the tornado, concern about fire risk to the nearby village prompted the Nature Conservancy to have a commercial logger remove all woody debris from a 15 m strip around the periphery of the damaged area.

### 2.2. Field and Statistical Methods

Canopy tree sampling was concentrated roughly in the center of the most severely-disturbed portion of Cathedral Pines, which is also the older section (the southern and eastern portions are much younger). An inventory of damage to pre-disturbance canopy trees was carried out in August 1991 in six 20-m by 20-m contiguous plots, in two rows of three. Within each plot, the following were recorded for all pre-disturbance trees > 10 cm diameter at breast height (dbh): species, status (living or dead), diameter at breast height (1.4 m above ground; dbh), type of treefall (snap vs. uproot vs. bent), and the long dimension of the root plate and pit for uprooted trees. Trees deemed dead at the time of the storm were excluded; those dead in 1991 had died since the 1989 storm. There were no obvious indications that the canopy tree composition and structure of the damage survey area were not representative of the slightly larger area covered by the revegetation survey (see below); therefore characteristics of the damaged survey area (e.g., basal area, biomass) were used as the pre-disturbance benchmark.

The recovering vegetation was sampled in August 1999, in 55 plots (2 m radius) distributed along five, 100-m long parallel transects. These transects began near the base of the hillside and continued up the face of the hill to the east. The area covered by the vegetation regeneration plots was centered on, but somewhat larger than, the area covered by the canopy tree damage survey; thus it encompassed some less-severely damaged areas. For all woody plants > 2 m tall and within 2 m of the plot center stake, the diameter at breast height (1.4 m; dbh), and species identity was recorded.

Species diversity was explored via rarefaction curves. Based on the 87 stems of pre-disturbance canopy trees in the six large plots, three rarefaction curves were constructed from (a) the pre-disturbance canopy trees (all 87 stems); (b) the 1999 survivors and regeneration stems > 5 cm dbh; and (c) the 1999 survivors and regeneration stems > 2 m tall. In the latter two cases, the rarefaction curves were calculated up to 87 stems, for comparison with the pre-disturbance trees. Rarefaction was accomplished with the iNextOnline soft-

ware (chao.shinyapps.io/iNextOnline/, accessed on 11 January 2021), set to generate 95% confidence limits.

A number of trees and saplings surveyed in 1999 were clearly too large to have been established after the tornado and were therefore classified as having been advanced regeneration at the time of the 1989 storm. The aboveground biomass in 1989 of these surviving stems was estimated as follows. Potential diameter growth rates by species were obtained from [30] and used to infer the 1989 dbh by subtracting the estimated annual diameter growth across each year, beginning with the 1999 dbh. The potential diameter growth rates derived from [30] were species- and size-specific, and therefore the amount subtracted from the "current" dbh changed slightly from year to year because of the change in stem diameter. Thus, for example, a *B. lenta* that was 6.8 cm dbh in 1999 would have a growth increment of 1.943 mm, resulting in a 1998 dbh of 6.61 cm. This tree would in turn have a 1998 growth increment of 1.964 mm, resulting in a 1997 dbh of 6.41, etc. Because of the nearly-complete canopy removal by the disturbance, the neighborhood competition term available in [30] was not used in the calculation of annual diameter growth.

Whether using the pre-disturbance tree dbh, the inferred 1989 dbh, or the measured 1999 dbh, biomass was calculated using the species-specific allometric equations of [31] for stems > 10 cm dbh, and of [32] for stems between 1 cm and 10 cm dbh. It was assumed that aboveground biomass was approximately 50% carbon, and therefore biomass was multiplied by 0.50 to obtain estimated carbon, which comprises the majority of what is reported here. Variability is estimated on the basis of six plots for the pre-disturbance estimates of basal area, biomass and carbon; and on the basis of 55 plots for the estimated 1989 and directly-measured 1999 results. Annual rates of accumulation assume linear change between 1989 and 1999 and are simply the difference between the biomass in 1989 and 1999, divided by ten.

## 3. Results

### 3.1. Pre-Disturbance Canopy Structure and Composition

Prior to the disturbance, the sampled area supported a basal area of 58.50 m²/ha of trees (>10 cm DBH), the majority of which was *T. canadensis* (45.50 m²/ha), with smaller amounts of *P. strobus* (5.09 m²/ha), *Quercus alba* (5.11 m²/ha), and other species (Figure 1).

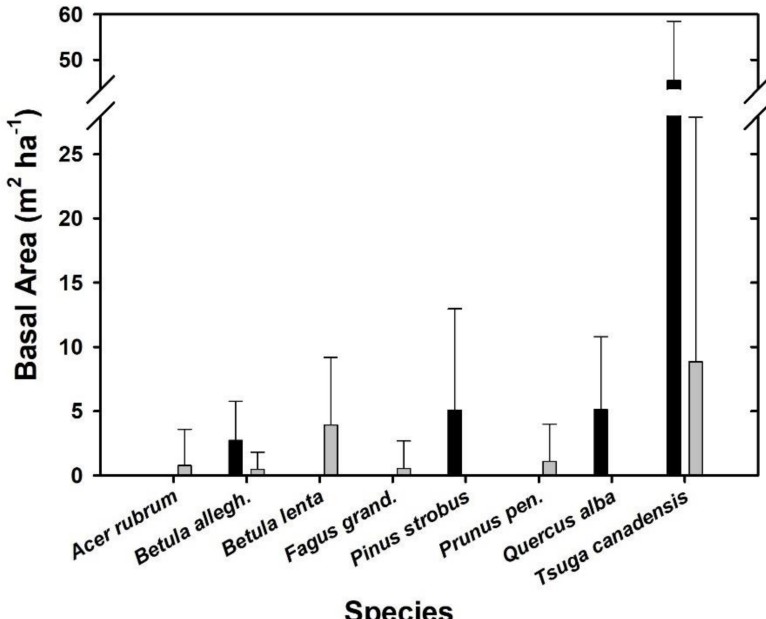

**Figure 1.** Pre-disturbance (black bars) and ten years post-disturbance (grey bars) basal area of standing live trees > 10 cm dbh, for the dominant species.

Pre-disturbance tree density (362 trees/ha) was dominated by *T. canadensis*, again with relatively similar minor contributions from other species. Individuals of only five species were of canopy size (>10 cm dbh) in the 0.24 ha area sampled (Figure 1). Trees of different species differed significantly in size (Kruskal-Wallis test, $\chi^2$ = 11.83, *p* = 0.019, 4 d.f.). *Q. alba* was, on average, the largest (72.0 $\pm$ 5.5 [mean $\pm$ std. dev.] cm dbh), followed by *P. strobus* (59.2 $\pm$ 22.6 cm dbh), and then *T. canadensis* (39.4 $\pm$ 18.1 cm dbh).

Total pre-disturbance aboveground carbon was quite high at 209.4 $\pm$ 50.5 Mg ha$^{-1}$ (Figure 2). Spatial variation of basal area and aboveground carbon was also high, with a roughly two-fold difference among the six 20 m $\times$ 20 m plots. Smaller, 100 m$^2$ grid cells reached maxima of 116.5 m$^2$ ha$^{-1}$ for basal area and 457.2 Mg ha$^{-1}$ for aboveground carbon.

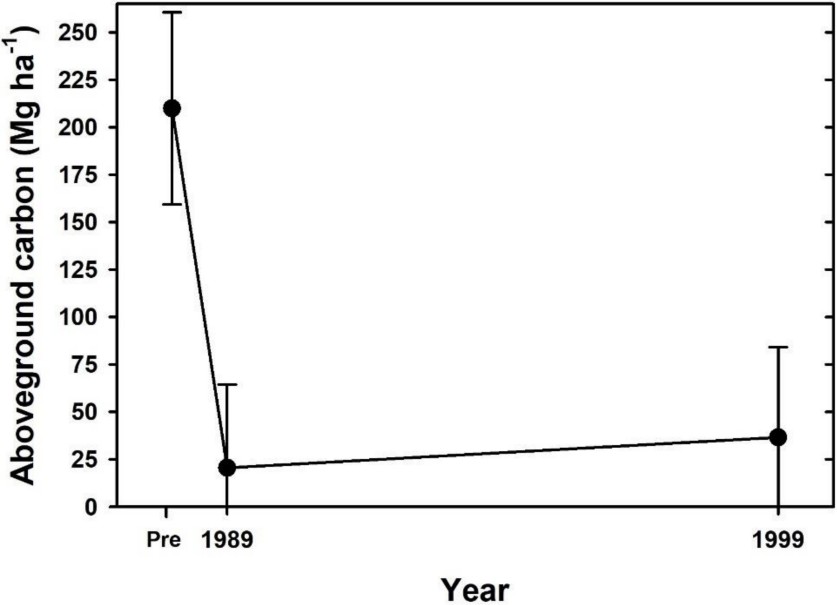

**Figure 2.** Aboveground live tree carbon from pre-disturbance, immediately after disturbance, and ten years after disturbance. Error bars are one standard deviation; pre-disturbance sampling from six 20 m $\times$ 20 m plots; 1989 and 1999 sampling from 55, 12.57 m$^2$ plots.

### 3.2. Structural Damage, Mortality, and Sprouting

The tornado reduced standing canopy tree density by 96% and basal area by 99% in the damage survey area, and while seedling mortality from the storm was not quantified, it appeared to be limited to the small percentage of seedlings literally crushed beneath fallen trees. Nevertheless, scattered saplings and small pole-size trees (most < 10 cm dbh) remained standing in the larger revegetation sampling area. The estimated 1989 post-disturbance aboveground carbon for the revegetation sampling area was 20.7 $\pm$ 43.9 Mg ha$^{-1}$, which was 9.9% of the pre-disturbance levels (Figure 2). Among the 87 trees (>10 cm dbh) in the damage survey area, 64 (74.4%) uprooted, 20 (23.2%) experienced trunk breakage (snapped), and 3 lost branches or were bent (2.4%).

The disturbance also greatly reduced the mean diameter of standing trees in the damage survey area, from 41.1 cm ($\pm$19.1) to 13.1 cm ($\pm$4.8) dbh; damage was clearly much greater in larger sizes.

Mortality was already very high by 1991; among the >10 cm dbh trees sampled, only 9.2% remained alive relative to the pre-disturbance tree density. Type of damage appeared to strongly influence survival in the 1991 survey; snapping and uprooting were clearly much more likely to cause mortality than lesser forms of damage.

Based on inferred sizes and abundances of advanced regeneration in 1989, and the few surviving canopy trees found in 1991, the woody species had 7.8 $\pm$ 14.2 m$^2$ ha$^{-1}$ of immediate post-disturbance basal area. The dominant species soon after the disturbance

was *T. canadensis* (Figure 1), with 67.9% of basal area. *B. lenta* was a distant second (19.4% of basal area).

### 3.3. Revegetation

A total of 363 saplings and trees were inventoried in 1999, of which 145 were estimated (via the size- and species-specific potential diameter growth rates from [30]) to have been ≥1 cm dbh in 1989, and 13 were estimated to have been >10 cm dbh in 1989. The basal area in 1999 for stems estimated to be post-disturbance recruits (all species pooled) was 9.58 m$^2$ ha$^{-1}$. The ranking for the relative basal area among post-disturbance recruits was *B. lenta* (62% of total), *T. canadensis* (32% of total), *P. pensylvanica* (17% of total), *A. rubrum* (12% of total), and *Fagus grandifolia* (9% of total). In terms of frequency, *B. lenta* was present in the highest proportion of sample plots.

Combining the post-disturbance recruits with the advanced regeneration that survived the disturbance, gives an overall picture of the 1999 vegetation. Ten years after the disturbance, the total basal area of surviving trees, released understory stems, sprouts, and new seedlings was 16.71 m$^2$ ha$^{-1}$ or 28.5% of the pre-disturbance basal area. *T. canadensis* was the dominant species in 1999, with twice the basal area (8.8 m$^2$ ha$^{-1}$) of the second most dominant species, *B. lenta* (3.9 m$^2$ ha$^{-1}$), as well as the greatest mean dbh among species (Figure 3). Other major species were *P. pensylvanica* (1.1 m$^2$ ha$^{-1}$), *A. rubrum* (0.8 m$^2$ ha$^{-1}$), and *B. alleghaniensis* (0.5 m$^2$ ha$^{-1}$). The *P. strobus* and *Q. alba* that were important in the pre-disturbance canopy were virtually absent.

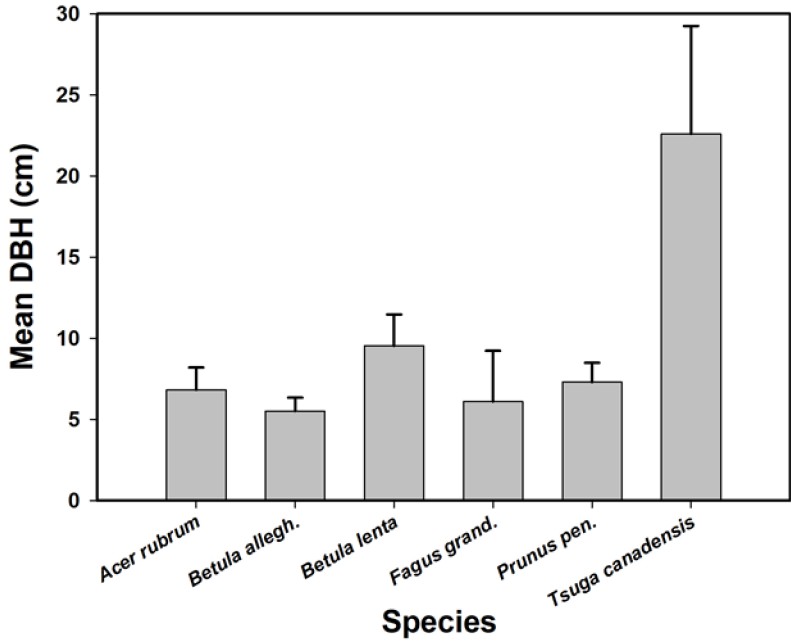

**Figure 3.** Mean and standard deviation of diameter at breast height (dbh, in cm) for stems in recovering stand, ten years after disturbance.

Species diversity among tree species was greater in 1999 than prior to the disturbance. Rarefaction to a total of 87 stems (the number of > 10 cm pre-disturbance trees in the large plots) shows (Figure 4) many more species for a given number of stems in the 1999 survey, whether counting all stems > 2 m tall, or only those saplings > 5 m dbh. The 1999 survey documented a total of 20 woody species > 2 m tall, and 11 woody species > 5 cm dbh. The lack of overlap of error bars shows that the higher diversity in both regeneration categories was significantly greater than in the pre-disturbance canopy trees.

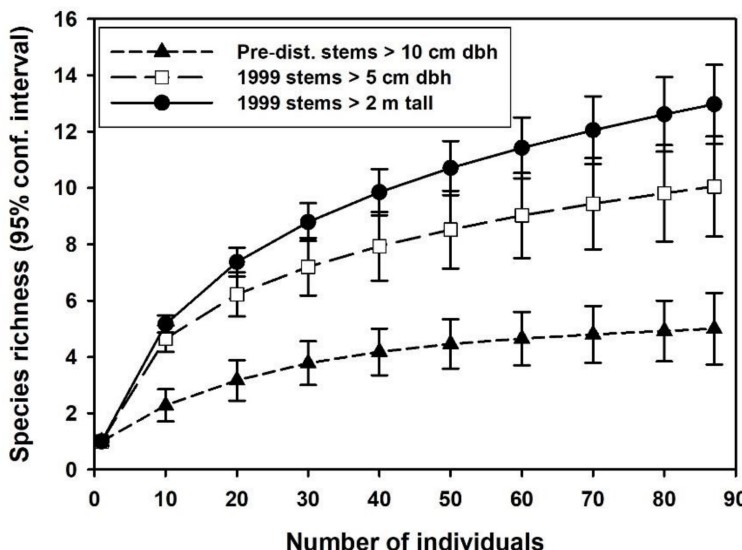

**Figure 4.** Species-accumulation curves for pre-disturbance canopy trees and two size classifications of post-disturbance woody vegetation in 1999 (combining survivors and new recruits). Error bars = 95% confidence intervals.

Aboveground live-tree carbon increased from $20.7 \pm 43.9$ Mg ha$^{-1}$ in 1989 to 37.1 Mg ha$^{-1}$ in 1999 or 17.7% of the pre-disturbance levels. This amounted to a mean accumulation rate of $1.6 \pm 0.7$ Mg ha$^{-1}$. This annual rate across 10 years was compared to the annual accumulation rate across six years at the Tionesta site [17]; non-normality of data and disparate sample sizes required a non-parametric Mann-Whitney rank sum test (Figure 5). The rate at Cathedral Pines (median = 1.6 Mg ha$^{-1}$) was significantly greater than the rate at Tionesta (median = 0.3 Mg ha$^{-1}$; U = 5288, T = 12,147, $n_{small}$ = 55, $n_{big}$ = 289, $p < 0.001$).

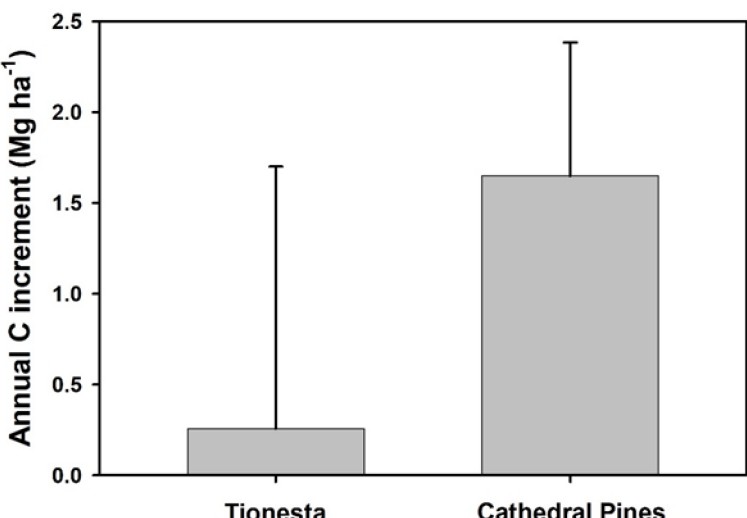

**Figure 5.** Annual carbon accumulation rates (Mg ha$^{-1}$) across six years at Tionesta (ref. [17], $n = 289$) and across ten years ($n = 55$) at Cathedral Pines. Bars show median rate; error bars indicate 75th percentile.

## 4. Discussion

Cathedral Pines, while not meeting strict definitions of "old-growth" in terms of stand history, nevertheless easily meets structural criteria for old-growth forests of the eastern

United States. [33] characterized old-growth forest remnants as having aboveground biomass of 220–260 Mg ha$^{-1}$ and up to 30% of aboveground biomass in trees > 70 cm dbh; the biomass values of 418 Mg ha$^{-1}$ and 19.4% at Cathedral Pines suggest that it was structurally similar to typical northeastern U.S. old-growth sites, see also [18,34].

The 1989 tornado that struck Cathedral Pines dramatically altered aboveground carbon in live trees, size distribution, and species diversity, while species composition showed small to moderate changes. This disturbance was unusually severe in comparison to the majority of wind disturbance studies [9,35], and the high severity of the disturbance, along with the particular species that were then dominant and the landscape setting, may provide useful context for understanding patterns of damage and recovery, see a similar concept in [7].

The pre-disturbance Cathedral Pines stand, comprised mostly of *Q. alba*, *P. strobus*, and *T. canadensis*, may have been especially vulnerable to wind damage [36]. The dominant conifers, *T. canadensis* and *P. strobus*, are not only species known to be vulnerable to windthrow [19,37], but were present as unusually large individuals. Larger trees are nearly always more vulnerable to wind than smaller ones e.g., [10,36]. In addition to vegetational influences, the thin, rocky soils on sometimes steep slopes at Cathedral Pines (personal observation) may have provided poor anchorage for the trees, and the exposed hillside provided no topographic sheltering from the winds of the tornado. Consequently, the canopy destruction at Cathedral Pines was especially severe.

Among the damaged pre-disturbance canopy trees, only 9.2% survived the first two years, a notable contrast to trends observed in the simulated hurricane created in 1990 at Harvard Forest [38], although the high survival observed there in years 1–5 declined substantially in subsequent years [11]. The limited survival and lack of resprouting of damaged individuals (personal observation) may be largely due to *T. canadensis* and *P. strobus*—two non-sprouting conifers—being dominant. In some wind-damaged tropical forests, the great majority of damaged but standing trees sprout [5], some sites in [7], providing a means for maintaining the species composition, as well as rapid regrowth because tall trunks and intact root systems already exist. To the extent that surviving trunks sprout, a wind-disturbed forest should therefore exhibit rapid regrowth and limited shift in species composition.

The moderate level of new seedling establishment at Cathedral Pines was largely made up of intolerant (*P. pensylvanica*) and mid-tolerant (*B. lenta*) species, but did substantially increase woody plant richness. Germination of their seeds may have been facilitated by the prevalence of uprooting rather than trunk breakage among canopy trees, which created a large number of treefall pits and mounds whose disturbed soil often prompts abundant seed germination of colonizing tree species [39,40]. As has been observed in other situations where a single species overwhelmingly dominates, events that break that dominance, even briefly, can facilitate a wave of establishment that augments diversity even if the pre-event dominant eventually reasserts itself. [41,42]. Nevertheless, although new seedling colonization was common, this regeneration mechanism was of secondary importance by the tenth year, and post-disturbance recruits were not destined to be the initial canopy dominants (Figure 6).

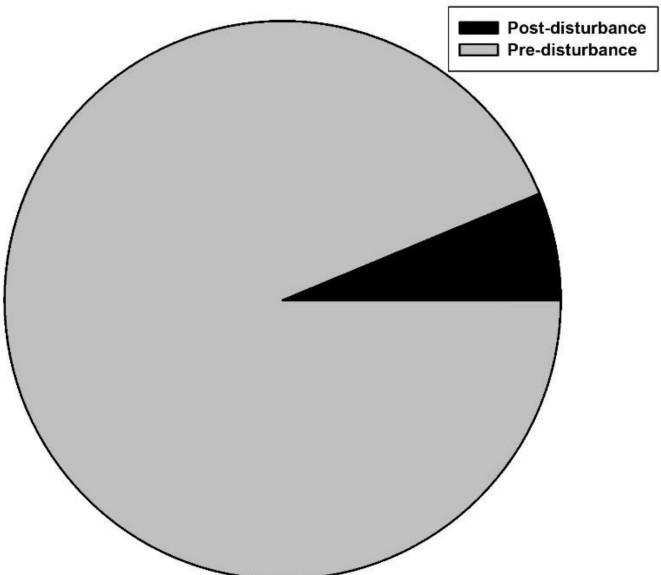

**Figure 6.** Proportion of 1999 basal area made up of stems of pre-disturbance vs. post-disturbance origin. Cohorts defined based on inferred 1989 dbh; all stems with inferred 1989 dbh > 1 cm were considered of pre-disturbance origin.

The majority of dominant stems ten years after the disturbance was derived from previously-suppressed *T. canadensis* saplings that were 1–10 cm dbh at the time of disturbance; this is consistent with trends observed in several other post-windthrow revegetation studies e.g., [14,19]. Closed-canopy *T. canadensis* stands, such as existed prior to the disturbance at Cathedral Pines, are very heavily shaded, which likely explains the paucity of species other than *T. canadensis* in the advanced regeneration [43,44]. Given the substantial size advantage of the *T. canadensis* at the time of disturbance, along with the potential long life of this species, this species might be expected to be canopy dominant for many decades or a century or more. In this scenario, the recovering forest in the first several decades would be a mix of released advanced regeneration (with *T. canadensis* dominant), with subordinate seedling recruits of several hardwood species. However, it is likely that *T. canadensis* will experience high mortality from the hemlock woolly adelgid, suggesting that the more likely species composition scenario is that *T. canadensis* will steadily decline in dominance over several decades, leaving *B. lenta* to inherit the dominant role, with *A. rubrum* and *P. serotina* as subordinates, and several other tree species contributing to elevated (compared to pre-storm) species diversity over the next century. The result is likely to be very little compositional change initially, which then steadily increases until the pre-disturbance dominant species is eliminated. Thus, the wave of post-disturbance new seedling establishment will likely eventually result in a more diverse forest, albeit one that initially passes through a low-diversity phase of dominance by *T. canadensis*. Figure 7 shows the compositional variation among the 1999 survey plots, as well as their distinctiveness from the pre-disturbance canopy trees; this figure thus hints at the more variable composition in the smaller seedling layers that will emerge after mortality of the initially dominant *T. canadensis*. The future more-diverse stand may even be less vulnerable to wind disturbance due to the absence of *P. strobus* and *T. canadensis*. Hypothesis 1 would therefore appear to be confirmed initially, but in later decades the increase in diversity and change in composition would suggest that Hypothesis 1 is no longer supported.

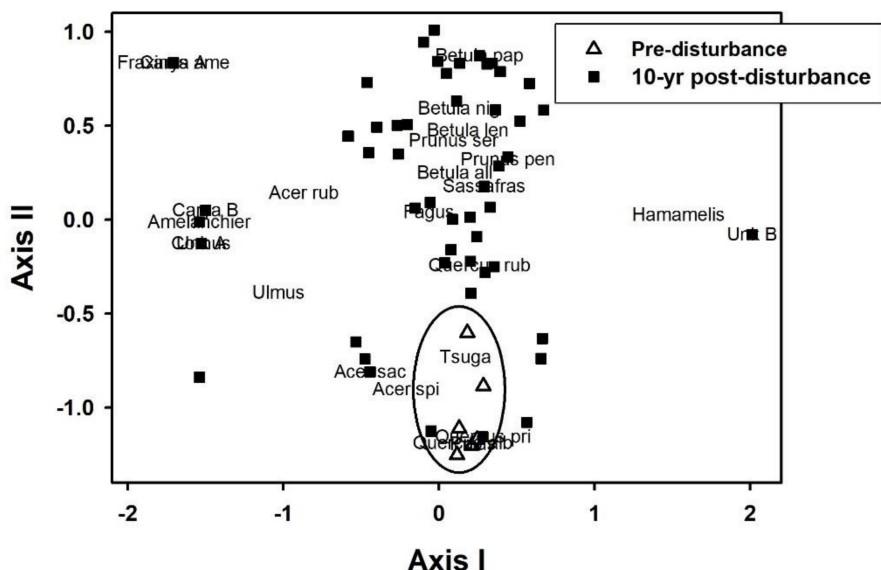

**Figure 7.** Nonmetric multidimensional scaling ordination of pre-disturbance canopy trees (open triangles) and 1999 seedlings and saplings (closed squares). Ellipse encloses all of the pre-disturbance sample plots. Positions of species in ordination space are also shown with text. Final stress for 2-dimensional solutions = 26.71.

This proposed temporal dynamic does not neatly fit the predictions of any of the major successional concepts, nor does it match empirical trends at two similar forests after catastrophic wind damage. First, the Cathedral Pines trajectory appears quite distinct from that in the old-growth Tionesta forest of western Pennsylvania [15,17], where the pre-disturbance dominant hemlock and beech will be replaced for at least the first century by a mix of *B. lenta* and *B. alleghaniensis*, with smaller amounts of *A. rubrum*. Hemlock at the Tionesta site was eliminated by pervasive deer browse combined with a summer drought that killed small seedlings in the third post-disturbance year. Beech advance regeneration was abundant immediately after disturbance at Tionesta, but only as small (i.e., <3 m tall) stems; these survived but were rapidly outgrown by *Betula* species in the first two decades, and are already being eliminated by beech bark disease (personal observation). Thus, the Tionesta site recovered mostly via seedling recruits and species composition changed substantially [15,17].

Second, the 1938 New England hurricane severely damaged a white pine-dominated landscape. Recovery across the first five decades [45,46] was a mix of released advance regeneration and post-disturbance seedling recruits. The composition was dominated by fast-growing but short-lived *P. pensylvanica* and *Betula populifolia* initially, but these species declined dramatically or disappeared by 40 years post-disturbance, allowing the remaining white pine to share dominance with a mix of longer-lived hardwoods. In contrast, Cathedral Pines will be dominated in the first decades after disturbance by the pre-disturbance dominant, which is likely to then be gradually eliminated by the impacts of an introduced insect, leaving somewhat long-lived birches and red maple as dominants.

The 1990 simulated hurricane experiment at Harvard Forest [11,38] provides a third comparison and is the site whose post-disturbance dynamics are most similar to Cathedral Pines. Severity was somewhat lower in the simulated hurricane experiment than at Cathedral Pines, and by 20 years after the disturbance, surviving canopy trees were the dominant mechanism of regeneration and oaks the largest contributor to the basal area. Nevertheless, the simulated hurricane also resulted in lesser but substantial contributions from new seedlings as well as released advanced regeneration. A striking compositional similarity between Cathedral Pines, Tionesta, and the simulated hurricane is that *B. lenta* became a dominant or subdominant species in all three situations; this species appears

likely to be the benefactor of wind disturbances across a large area of Mid-Atlantic and New England portions of the U.S.

Thus, the oft-mentioned tendency for wind disturbances to result in little compositional change is probably restricted to events that allow survival and release of subcanopy and sapling-size stems that are a similar species to the pre-disturbance canopy. Such would be the scenario for Cathedral Pines in the absence of the likely upcoming *T. canadensis* mortality from the adelgid. Wind disturbances are likely to produce substantial changes in the composition if abundant new seedling establishment is favored [47], such as when severity is very high and chronic herbivory has eliminated the advanced regeneration layer [15].

Aboveground live-tree carbon accumulation rates fit expectations for very severe disturbance of conifer forests where the release of large advanced regeneration occurs. Because damaged *T. canadensis* and *P. strobus* cannot recover by sprouting, that mechanism of regeneration was largely precluded at Cathedral Pines, reducing the maximal possible recovery rate, see [5]. Conversely, growth of released large advance regeneration stems may allow more rapid aboveground carbon accumulation than situations where recovery must start from very small new seedlings [15,17]; this is shown in the significantly higher carbon accumulation at Cathedral Pines than at the Tionesta site, leading to confirmation of Hypothesis 2. Together, these two factors seem to have resulted in an intermediate rate of accumulation during the first post-disturbance decade at Cathedral Pines; the rate of $1.6$ Mg ha$^{-1}$ yr$^{-1}$ found here is intermediate among the comparable temperate studies, but well below carbon accumulation rates reported for some tropical sites.

This study was originally conceived to document compositional and species diversity changes rather than biomass and carbon dynamics, and as such, has some limitations in regards to the latter purposes. In light of the finding in [17] that carbon accumulation was slow in the first few years but then accelerated, this study would have been improved with additional sample dates between 1991 and 1999. The objective of the 1991 sampling was to document canopy tree damage, therefore direct information is lacking on seedlings and saplings at the time of the disturbance or shortly thereafter. The lack of such data may have substantially compromised findings if regeneration had relied mostly on seedling or small sapling recruitment, as in [17]. However, in this case, the large saplings and subcanopy trees at Cathedral Pines were released by the disturbance and will dominate in the early decades of regeneration; these early dominants are almost entirely *T. canadensis*, which casts especially deep shade, which will preclude recruitment from below as long as the *T. canadensis* dominants maintain a closed canopy. In terms of biomass and carbon, seedlings < 2 m tall are a very small fraction of total biomass and carbon, so the totals reported here are an underestimate, although a very small one. Lastly, the 1991 and 1999 sampling was conducted with different numbers and sizes of plots. The small number (6) of large plots in 1991 was used because of the focus on canopy trees, whereas the larger number (55) of small plots in 1999 was used to focus on smaller seedlings and saplings. The small plots were broadly and approximately evenly spread throughout the area covered by the 1991 sampling but the slightly larger area covered in 1999 did include some less severely-disturbed zones. This likely led to slightly higher estimates of the 1989 basal area and aboveground carbon totals than if the small plots had been entirely restricted to the most severely damaged central zone.

## 5. Conclusions

Several aspects of the damage and regeneration of the Cathedral Pines forest follow trends observed in a few other wind-disturbed old-growth forests in which *P. strobus* and/or *T. canadensis* were dominants [17,18]. It is suggested that high severity and the species identity of damaged trees, combined to minimize tree survival and sprouting and to create abundant disturbed-soil microsites that facilitated the seedling establishment of early-seral hardwood species. Compared to a forest of similar composition that experienced similar severity of disturbance, the biomass and carbon accumulation rate here was higher,

as hypothesized for sites where regeneration derives primarily from the release of advanced regeneration and suppressed saplings.

**Funding:** Initial sampling was carried out with funding from the Nature Conservancy Small Grants program. During part of this research period, I was supported by the National Science Foundation grant BSR 91-07243. The 1999 resurvey was funded by a grant from the Andrew Mellon Foundation.

**Data Availability Statement:** The data presented in this study are openly available in Figshare at https://doi.org/10.6084/m9.figshare.c.4926801.v1 (accessed on 11 January 2021), reference "Cathedral Pines 10-year recovery".

**Acknowledgments:** I thank the Connecticut chapter of The Nature Conservancy for permission to conduct this research at Cathedral Pines. I particularly acknowledge Julie Tritschler for setting up plots, and Scott Robison and Bob Duff for assisting with the damage and revegetation surveys.

**Conflicts of Interest:** The author declares no conflict of interest.

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
