# Peer review of "First-Decade Biomass and Carbon Accumulation, and Woody Community Change after Severe Wind Damage in a Hemlock-White Pine Forest Remnant"

_forests, doi:10.3390/f12020231_

Round 1

Reviewer 1 Report

Lines 17-19/  This study augments the limited number of longer-term empirical studies that report biomass and carbon accumulation rates after wind disturbance, and can therefore serve as an benchmark for mechanistic and modelling  research. 

The site is a hemlock-white pine forest in northwest Connecticut with old-growth characteristics.  About half of the site (8 hectares out of the 16.2 ha) was severely damaged by a tornado.  Pre-disturbance biomass and carbon was estimated based on the characteristics of the part of the location that was not damaged, and measurements were also taken ten years later for trend data.

This kind of “benchmarking” information is useful, and the analyses is solid.  The experimental design and plot protocols are somewhat ad hoc looking, and it could be easy to criticize those, but it is understandable given the difficulty in studying this type of disturbance.  The text is quite clear as to the methods, which is important.

One aspect that needs improvement is to clarify the proposed hypotheses. Although the abstract focuses on the benchmarking aspect of this study for a severe wind disturbance, lines 67-73 in the introduction proposes hypotheses (expectations) that could be more clearly phrased as hypotheses, and the results of testing the hypotheses should be included in the abstract.   The hypotheses in L67-72 need to be more generally written, because the site description does not occur until section 2.  It should not be that difficult to generalize the hypotheses such that no matter what the characteristics of the site (in terms of subcanopy and sapling layer for example), results would be presented so that the hypothesis could be tested about expected carbon accumulation rates.  The results of testing the hypotheses (expectations) seem to be addressed in lines 307-308, but explicitly addressing the results more clearly is needed.   Addressing this aspect should only take a small amount of rewriting.

Author Response

Please see attached response to referees file.

Reviewer 2 Report

My strong belief is that the goal of the study in our subject is to gain new knowledge that can improve the understanding of the processes in the nature, in this case – natural succession after sever windthrow. I was not convinced on that by reading current paper. Destruction of the studied site by tornado took place in year 1989 while the monitoring of the regeneration and growth (carbon accumulation) was performed in year 1999. If the study object is still available for measurements – what is a rationale to operate with 20-year-old data? There are plenty of studies on the subject where researchers have analyzed the factors influencing regeneration or growth rates of established trees after wind throws. In current paper author has limited his work on description of situation not trying to apply any statistical method to explain the results. The interpretation of obtained data is based on previous experience not providing any proof by data analysis. In general, the article is written in good language, however, the aim of the study in not clearly defined and conclusions are vague. I have no doubt that author is an expert in subject area what is reflected in his previous publications (e.g. https://doi.org/10.1016/S0378-1127(00)00283-8). Unfortunately I have to say that current submission is not matching previous standards.

Author Response

(The authors gave the same response as above.)

Round 2

Reviewer 2 Report

The author has greatly improved his submission after revision.  In general this is a good quality paper.